# Genome-Wide Identification and Expression Pattern Analysis of *TIFY* Family Genes Reveal Their Potential Roles in *Phalaenopsis aphrodite* Flower Opening

**DOI:** 10.3390/ijms25105422

**Published:** 2024-05-16

**Authors:** Yunxiao Guan, Qiaoyu Zhang, Minghe Li, Junwen Zhai, Shasha Wu, Sagheer Ahmad, Siren Lan, Donghui Peng, Zhong-Jian Liu

**Affiliations:** Key Laboratory of National Forestry and Grassland Administration for Orchid Conservation and Utilization, College of Landscape Architecture and Art, Fujian Agriculture and Forestry University, Shangxiadian Road No. 15, Cangshan District, Fuzhou 350002, China; guanyunxiao@fafu.edu.cn (Y.G.); 52319026087@fafu.edu.cn (Q.Z.); fjalmh@fafu.edu.cn (M.L.); zhaijunwen@fafu.edu.cn (J.Z.); shashawu1984@126.com (S.W.); sagheerhortii@gmail.com (S.A.); lkzx@fafu.edu.cn (S.L.)

**Keywords:** genome-wide analysis, *Phalaenopsis*, TIFY gene family, flower opening, expression analysis

## Abstract

The TIFY gene family (formerly known as the zinc finger proteins expressed in inflorescence meristem (ZIM) family) not only functions in plant defense responses but also are widely involved in regulating plant growth and development. However, the identification and functional analysis of TIFY proteins remain unexplored in Orchidaceae. Here, we identified 19 putative *TIFY* genes in the *Phalaenopsis aphrodite* genome. The phylogenetic tree classified them into four subfamilies: 14 members from JAZ, 3 members from ZML, and 1 each from PPD and TIFY. Sequence analysis revealed that all *Phalaenopsis* TIFY proteins contained a TIFY domain. Exon–intron analysis showed that the intron number and length of *Phalaenopsis TIFY* genes varied, whereas the same subfamily and subgroup genes had similar exon or intron numbers and distributions. The most abundant *cis*-elements in the promoter regions of the 19 *TIFY* genes were associated with light responsiveness, followed by MeJA and ABA, indicating their potential regulation by light and phytohormones. The 13 candidate *TIFY* genes screened from the transcriptome data exhibited two types of expression trends, suggesting their different roles in cell proliferation and cell expansion of floral organ growth during *Phalaenopsis* flower opening. Overall, this study serves as a background for investigating the underlying roles of *TIFY* genes in floral organ growth in *Phalaenopsis*.

## 1. Introduction

TIFY is a plant-specific gene family, formerly known as the zinc finger proteins expressed in inflorescence meristem (ZIM) family, which was discovered in *Arabidopsis* [1]. It was later renamed the TIFY gene family because of its highly conserved TIFY domain, which contains the TIF [F/Y] XG amino acid sequence [2]. TIFY proteins are divided into four subfamilies based on the typical types of their additional domains: TIFY, JAZ (jasmonate-ZIM-domain), PPD (PEAPOD), and ZML (ZIM/ZIM-like) [3]. Among these, the TIFY subfamily possesses only one conserved TIFY domain. In addition to the TIFY domain, the PPD subfamily contains a unique N-terminal PPD domain. The JAZ subfamily includes one CCT_2 structural domain, and the ZIM subfamily harbors one ZnF_GATA domain [3].

Flowers are the reproductive organs of angiosperms and also a crucial reflection of ornamental traits and economic value for landscape plants. Therefore, research on flower development has gained considerable attention from horticulturists to create more diverse flower patterns to meet the various tastes of different consumers. A typical orchid plant flower contains four parts: three outer sepals, two inner petals, one specialized lip, and a reproductive organ called a column [4]. Numerous studies have been conducted on floral organ identity, and specific flower development models for orchids have been proposed [5,6,7]. Nevertheless, little is known about the genetic regulation of the late stage of flower development in orchids. This determines the specific growth patterns of the final flower organ size and flower senescence.

To date, numerous studies have confirmed that TIFY proteins are implicated in various processes of plant growth and development, such as root growth [8], the formation of the top hook of seedlings [9], trichome initiation [10], leaf senescence [11], tuber initiation and bulking [12], fuzz fiber yield [13], seed fertility [14], stamen development [14], anthocyanin accumulation [10], spikelet development [15], and flowering regulation [16]. Furthermore, Guan et al. [17] identified a TIFY family gene, *CmJAZ1-like*, most abundantly transcribed at the budding stage and gradually decreased with flower opening. They further confirmed that CmJAZ1-like can downregulate cell expansion-related genes by interacting with CmBPE2, thereby limiting the size of chrysanthemum petals. Oh et al. [18] generated an RNA interference transgenic plant, irJAZd, and proved that *NaJAZd* could retard flower abscission in *Nicotiana attenuata*. In roses, *RhJAZ5* transcripts were enriched at stage 5 (fully opened flowers with visible anthers), and the phenotypes of *RhJAZ5*-silenced and *RhJAZ5*-overexpressed plants demonstrated that *RhJAZ5* counteracted flower senescence [19]. These findings indicate that the *TIFY* family genes regulate the late stage of flower development.

*Phalaenopsis* has become one of the most traded flower crops in the world because of its elegant flower posture, rich color, long flowering period, and high economic value. The TIFY family of genes functions in various essential biological processes, including its vital roles in flower development. However, *TIFY* genes have not been reported in Orchidaceae, except *Dendrobium officinale* [20]. Therefore, it is important to identify and elucidate their potential roles in *Phalaenopsis*. Our study identified and investigated TIFY family genes based on *Phalaenopsis aphrodite* genome data. Phylogenetics, gene structures, conserved domains, synteny and evolutionary relationships, promoter *cis*-elements, and expression characteristics at the four stages of flower opening were analyzed. Our results provide clues for the further exploration of the evolutionary and biological functions of TIFY proteins in *Phalaenopsis*.

## 2. Results

### 2.1. Identification and Characterization of TIFY Genes in Phalaenopsis

Overall, the 14 JAZ, 1 PPD, 3 ZML, and 1 TIFY sequences were identified in the *P. aphrodite* genome. The full-length protein sequences are listed in Appendix A. As shown in Table 1, these genes were designated based on their homologues to *Arabidopsis* and their chromosomal positions: *PaJAZ1*-*14*, *PaPPD*, *PaZML1*-*3*, and *PaTIFY*. The ORF lengths of the 19 *TIFY* genes ranged from 339 (*PaJAZ13*) to 1188 bp (*PaTIFY*), and the respective numbers of amino acids (aa) ranged from 113 to 396 aa. Correspondingly, the predicted molecular weight (Mw) varied from 12.74 to 41.7 KDa, which was consistent with the ORF length. Moreover, the hydrophilic (GRAVY) values of all 19 TIFY proteins were negative, indicating strong hydrophilicity. Except for 4 genes (*PaJAZ9*, *PaJAZ10*, *PaZML2*, and *PaZML3*), 15 of 19 *TIFYs* in *Phalaenopsis* showed theoretical isoelectric point (pI) values higher than 7.0, revealing that most of them were alkaline. The instability index of the 17 TIFY proteins was greater than 40, suggesting their unstable nature [21]. Subcellular localization predictions showed that all TIFY proteins were located in the nucleus, implying that they may function like transcription factors or form interacting protein pairs with TFs (Table 1).

### 2.2. Phylogenetic Analyses and Classification of the Phalaenopsis TIFY Gene Family

To analyze the classification and evolutionary relationships of the identified TIFY proteins in *Phalaenopsis*, a phylogenetic tree was generated based on the alignment of 57 TIFY protein sequences, including 19 TIFYs from *P. aphrodite*, 20 TIFYs from *Oryza sativa*, and 18 TIFYs from *Arabidopsis thaliana* (Figure 1). The results showed that all 57 TIFYs were classified into four clades: JAZ, ZML, TIFY, and PPD. Among these, 41 TIFY members belonged to the JAZ subfamily and were separated into six subgroups (JAZ I to JAZ VI). JAZ I, JAZ II, and JAZ III covered the JAZ proteins from the three species; JAZ V subgroups only contained the JAZ proteins from *Phalaenopsis* and *Arabidopsis*; the JAZ IV subgroup only contained seven *O. sativa* JAZ proteins; and no AtTIFY was classified in the JAZ VI subgroups. Similar findings have been reported in previous studies [22,23,24], indicating that JAZ VI might be unique to monocots.

The ZML subfamily contains four members from *O. sativa*, three from *Arabidopsis*, and three from *Phalaenopsis*. PaTIFY (PAXXG116340), OsTIFY (Os02g49970), and AtTIFY (AT4G32570) were clustered together in the TIFY subfamily. Previous studies have shown that the PPD protein is mostly found in dicot species [3,25] and is involved in coordinating tissue growth and regulating lamina size and leaf blade curvature [26]. The TIFY protein PaPPD (PAXXG088150) from *Phalaenopsis* was grouped into the PPD subfamily. These results indicate that PPD also exists in monocots; however, its function needs further exploration.

### 2.3. Sequence Analysis of Phalaenopsis TIFY Family

To further explore the structural and functional characteristics of *Phalaenopsis* TIFY proteins, the conserved motifs of 19 TIFY proteins were analyzed using the online tool MEME. Ten motifs were assigned as the upper bound, and the detailed sequences and logos are shown in Appendix A. As shown in Figure 2A, the number of motifs in *Phalaenopsis* TIFY proteins ranged from three to six. Among the ten conserved motifs, motifs two and seven were present in all *Phalaenopsis* TIFY proteins. Furthermore, some motifs existed only in a specific group, such as motif three, which only appeared in the JAZ IV and JAZ V subgroups, and motif six, which only existed in the ZML subfamily. TIFY proteins belonging to the same subfamily or subgroup generally contain the same type and number of motifs, and their distribution patterns are similar.

Intron–exon structure analysis was conducted to explore the structural diversity of *Phalaenopsis TIFY* genes. As shown in Figure 2B, the number of exons in these *TIFY* genes varied from one to eight. *PaJAZ13* had only one exon, and *PaPPD* had eight exons. Moreover, the first intron of *PaTIFY* was the longest among all sequences. The intron distributions also revealed that the same subfamily or subgroup of genes usually had similar exon and intron numbers and structures (Figure 2B). For example, the genes belonging to subgroups JAZ IV and V contained five exons, three JAZ II subgroup genes possessed seven exons, and four JAZ I subgroup genes, except for *PaJAZ13,* harbored three exons. In the ZML group, all three genes contained seven exons (Figure 2B).

In addition, the NCBI Batch CD Search Tool (https://www.ncbi.nlm.nih.gov/Structure/bwrpsb/bwrpsb.cgi, accessed on 8 December 2023) was used to examine the conserved domains of *Phalaenopsis* TIFY proteins and four conserved domains were identified: TIFY, Jas, CCT, and GATA (Figure 3A). All the TIFY family genes contain a conserved TIFY domain [3]. Indeed, the 19 *Phalaenopsis* TIFY proteins exhibited similar characteristics. Moreover, except for PaJAZ2, all JAZ and PPD subfamily proteins contained TIFY and Jas domains. The ZML subfamily proteins possess TIFY, CCT, and GATA domains. The TIFY subfamily protein, PaTIFY, contains only the TIFY domain. Subsequently, multiple sequence alignment was conducted using DNAMAN v9.0 software to explore the 19 TIFY domains’ conservation patterns further (Figure 3B). The results indicated that the TIFY domains of the 19 *Phalaenopsis* TIFY proteins were not completely conserved, but most contained common motifs. For instance, the TIFYXG motif is shared by most JAZ subfamily members (including JAZ II, JAZ III, JAZ IV, and JAZ V subgroups) and PPD subfamily members. The KIRYXV motifs specifically belong to the ZML family. The TIFY subfamily includes only one member, and its amino acid sequence in the TIFY domain is TVFYAG. However, the JAZ I subgroup’s TIFY domains (TMLYGG, SFFYGG, TIFYNG, and TIFING) are poorly conserved. In general, the gene structure and conserved domain composition supported the phylogenetic analysis described above, which might also be the main reason for the differentiation of gene function.

### 2.4. Promoter cis-Elements Analysis of Phalaenopsis TIFY Genes

Promoters are crucial factors that determine gene expression at the transcriptional level, and their regulation mainly relies on *cis*-acting elements located upstream of the genes. Therefore, the 2000 bp upstream regions of *PaTIFYs* were extracted for their putative *cis*-elements identification. We identified 442 *cis*-acting elements, including 36 types and 14 response functions (Figure 4; Appendix A). Various types and numbers of elements generate a wide range of gene functions. Among these elements, light responsiveness elements (G-box) were the most common (14.7%), followed by hormone-responsive elements (ABRE) (12.4%), which supported the idea that most biological processes in plants rely on light and hormone regulation. Furthermore, *PaJAZ7* had the highest number of 42 *cis*-acting elements, indicating a comprehensive function (Figure 4B).

Some *cis*-elements functioned in multiple phytohormone responsiveness, including methyl jasmonate (MeJA), abscisic acid (ABA), gibberellin (GA), auxin, and salicylic acid. Others are involved in stress responsiveness, such as anaerobic, low temperature, drought, anoxic, and wounding. In addition, there were elements associated with plant growth and development, such as circadian control, light response, and meristem expression (Figure 4A). Each *TIFY* gene had multiple elements involved in light responsiveness, which was the most prevalent function, implying that light mediates the function of *Phalaenopsis TIFYs* during plant growth and development (Figure 4). The *cis*-elements related to MeJA and ABA responsiveness were the second- and third-most abundant types, respectively, suggesting that these two phytohormones might regulate the transcriptional expression levels of *Phalaenopsis TIFY* genes.

### 2.5. Synteny and Evolutionary Analyses

First, we performed an intraspecific gene duplication analysis for the 19 *Phalaenopsis TIFY* genes. Nineteen genes were unevenly distributed across 16 linkage groups (LGs) of *P. aphrodite*, and four segmental duplication events were identified using the MCScanx method. These events included *PaJAZ7* on LG6 and *PaJAZ1* on LG1, *PaJAZ7* on LG6 and *PaJAZ4* on LG2, *PaJAZ5* on LG4 and *PaJAZ1* on LG1, and *PaJAZ4* on LG2 and *PaJAZ1* on LG1 (Figure 5A). Moreover, these segmentally duplicated genes were clustered in the JAZ IV and V subgroups, exhibiting similar conserved motifs and distributions (Figure 2A). To examine the potential selective pressure of the four segmentally duplicated gene pairs, the *Ka/Ks* ratios were obtained, and the results indicated that all *Ka/Ks* values were less than one (Appendix A), implying that the four pairs of *TIFY* genes in *Phalaenopsis* had suffered intense purification selection pressure.

Subsequently, comparative syntenic analyses of *Phalaenopsis* and two representative plant species (*A. thaliana* and *O. sativa*) were conducted. The results indicated that *Phalaenopsis TIFY* genes 15 with rice and 6 with *Arabidopsis* showed syntenic relationships. Five collinear gene pairs were found both in rice and *Arabidopsis*; one was unique to *Arabidopsis*, and nine were just found in the monocot rice, *O. sativa* (Figure 5B). These results suggest that *Phalaenopsis* and rice are monocots and share a relatively close evolutionary relationship.

### 2.6. Expression Analysis of TIFY Genes in the Process of Phalaenopsis Flower Opening

To investigate whether the *TIFY* family genes are involved in flower opening in *Phalaenopsis*, the expression profiles of 19 *TIFY* genes in small buds, large buds, and fully open flowers were mined based on the transcriptome data of *P. aphrodite* [27]. The results showed that 13 of the 19 *TIFY* genes exhibited relatively high expression levels during flower opening, suggesting a crucial role in this process (Appendix A). To further explore the changes in these 13 *TIFY* genes during flower opening, the late flower development process of *Phalaenopsis* was divided into four stages (S1–S4) based on the degree of flower opening (Figure 6A), and the whole flowering plant process is provided in Appendix A.

Next, qRT-PCR experiments were conducted to detect the expression levels of 13 *TIFY* genes mentioned above during the S1–S4 stages (Figure 6B). Interestingly, eight JAZ subfamily genes (*PaJAZ1*, *PaJAZ6*, *PaJAZ7*, *PaJAZ8*, *PaJAZ9*, *PaJAZ11*, *PaJAZ12*, and *PaJAZ14*) all showed an increasing trend from the S1 to S2 stage (peaking at the S2 stage), followed by a decreasing tendency from the S2 to S4 stage, with relative expression levels reaching their lowest level, except for *PaJAZ12*, which was equal to the S2 stage. These results suggest that the S2 stage is a crucial node for flower opening and that JAZ subfamily genes may play an important role in this phase. However, the expression levels of the remaining five *TIFY* genes (*PaTIFY*, *PaPPD*, *PaZML1*, *PaZML2*, and *PaZML3*) decreased from S1 to S4, indicating that the TIFY proteins which belong to the TIFY, PPD, and ZML subfamilies in *Phalaenopsis* may negatively regulate flower opening.

### 2.7. Expression Analysis of Petal- or Lip-Associated TIFY Genes during Flower Opening in Phalaenopsis

A complete *Phalaenopsis* flower comprises sepals, petals, lips, and a gynostemium. Moreover, the three types of perianth organs predominantly determined the ornamental characteristics. To further explore whether these 13 *Phalaenopsis TIFY* genes function in the growth of sepals, petals, and lips during flower opening, we first analyzed their tissue specificity based on published transcriptome data (Figure 7A). The results indicated that four *TIFY* genes (*PaJAZ1*, *PaJAZ7*, *PaJAZ9*, *PaJAZ11*, and *PaZML3*) were most abundantly transcribed in the petals, two (*PaPPD* and *PaZML1*) were highly expressed in the lips, and no *TIFY* gene was enriched in the sepals. These results indicated that the *TIFY* genes in *Phalaenopsis* might not play a significant role in sepal development.

The expression patterns of these petal- or lip-associated *TIFY* genes were investigated using qRT-PCR (Figure 7B). The results showed that *PaJAZ9* and *PaJAZ11* significantly reduced from the S1 to S4 stages, indicating their function as repressors of petal expansion growth. *PaJAZ1*, *PaJAZ7*, and *PaZML3* maintained high expression levels during the S2 and S3 stages, revealing that they might play a crucial role in petal cell expansion to push the petals out of the buds. In addition, the two lip-specific transcriptional genes exhibited different expression trends during the four stages. The relative expression of *PaPPD* peaked at the S1 stage and then decreased significantly. *PaZML1* was transcribed uniformly during the S1 and S2 stages and then increased and reached its highest level at the S3 stage. Moreover, these petal- and lip-associated *TIFY* genes exhibited lower expression levels at the S4 stage, indicating that they may not prevent flower abscission. Overall, these expression data provide clues for verifying the function of the *TIFY* gene in petal or lip development.

## 3. Discussion

TIFYs have attracted considerable attention as an essential gene family in plants. However, reports on TIFY proteins in Orchidaceae have been limited to *D. officinale*. Hu et al. [20] identified 20 *TIFY* genes in the genome of *D. officinale*. Among these, 11 *TIFY* genes were differentially expressed during the five stages of *D. officinale* protocorm development, suggesting that these genes may be involved in the development of *D. officinale* protocorm. We identified 19 *TIFY* family genes in *Phalaenopsis*, and subcellular localization predictions indicated that these proteins were all located in the nucleus (Table 1). In addition, the phylogenetic tree classified the 57 TIFY proteins (sourced from *Phalaenopsis*, rice, and *Arabidopsis*) into four subfamilies: JAZ, ZML, TIFY, and PPD. The JAZ subfamily is divided into six subgroups (JAZ I–VI) (Figure 1). Notably, the JAZ VI subgroup did not contain the *TIFY* family genes of *Arabidopsis* (Figure 1), and the same phenomenon has been observed in *Brachypodium distachyon*, *Phyllostachys edulis*, and *D. officinale* [20,23,24], revealing that this subgroup might be peculiar to monocots. The PPD subfamily is widely present in dicots plants [28,29,30,31], while for monocots, it has been identified only in *Tartary Buckwheat*, *D. officinale*, and *Phalaenopsis*.

The *TIFY* genes of *Phalaenopsis* exhibit significant variations in their sequence structure. The amino acid sequence length ranged from 113 to 396 aa (Table 1), the number of introns varied from one to eight, and the number of conserved motifs ranged from three to six (Figure 2). Some conserved motifs and domains were unique to specific genes or subgroups (Figure 2 and Figure 3). The large variation in sequence structure suggests that the *TIFY* family genes have changed their genome through evolutionary events, leading to functional differentiation [22,29]. Furthermore, the regularity of gene structure provides evidence for phylogenetic clustering.

Upstream transcription factors regulate gene expression levels, and the promoter region is crucial for this process [32]. Therefore, analyzing *cis*-acting elements in the promoters of the *Phalaenopsis TIFY* genes is an effective way to predict their functions. This study identified 14 functional types based on the regulatory elements in *Phalaenopsis TIFYs*’ promoter regions. These 14 function types mainly included phytohormone responsiveness, stress responsiveness, and the regulation of growth and development (Figure 4A). The promoter of each *TIFY* gene contained many *cis*-elements implicated in light responsiveness (Figure 4), indicating that environmental changes induced by light may affect the function of *TIFY* genes in *Phalaenopsis*. In addition, many *cis*-acting elements are involved in phytohormones, such as MeJA, ABA, GA, auxin, and SA (Figure 4). Indeed, certain phytohormones play important roles in the late stages of flower development. In *Arabidopsis*, jasmonate induces the accumulation of the *BIGPETAL* gene at the post-transcriptional level. This gene limits the post-mitotic cell expansion of the petals [33]. Moreover, GA and ABA regulate petal cell expansion in a mutually antagonistic manner in *Gerbera hybrida*, and auxins may also mediate this process [34,35]. The various functions of *cis*-elements indicate the multiple roles of *TIFY* genes in plant growth, and whether *TIFY* genes in *Phalaenopsis* are involved in the phytohormone-mediated flower-opening process requires further investigation.

Segmental gene duplication is the dominant factor for generating and maintaining gene families and is also considered the main source of gene structural changes and innovation [36]. The present study identified four segmentally duplicated gene pairs in the *Phalaenopsis* TIFY family (Figure 5A). One pair, PaJAZ1/PaJAZ7, showed similar expression patterns in the whole flowers during the four stages of flower opening (Figure 6). Furthermore, they were all transcriptionally abundant in the petals, especially during the S2 and S3 stages (Figure 7), suggesting that these two genes may perform vital functions in petal expansion. Syntenic analyses revealed that the number of collinear gene pairs between *Phalaenopsis* and rice was significantly higher than between *Phalaenopsis* and *Arabidopsis* (Figure 5B), demonstrating the distance of their genetic relationship.

*Phalaenopsis* is a widely traded flower crop with commercially valuable and charming flower types, and its long flowering period is a meaningful question researchers wish to address and introduce to other ornamental plants. In addition, breeding *Phalaenopsis* with different flower sizes is also a hotspot for horticulturists, which is very essential to cater to different consumers and markets. These two vital traits are closely related to flower opening. Therefore, we analyzed the expression trends of 13 *TIFY* genes selected from the transcriptome data at the four stages of flower opening. Overall, the changes in their expression levels exhibited two trends: eight *TIFY* genes showed an initial increase, followed by a continuous decrease (peaking at the S2 stage), and five *TIFY* genes displayed a persistent decline from the S1 to S4 stages (Figure 6). Cell proliferation and expansion jointly determine the opening process of flowers and the final size of flower organs. Floral organ primordia initially undergo cell proliferation followed by cell expansion to achieve a specific size [37]. Therefore, the group of genes highly expressed at the S2 stage may positively regulate cell expansion and promote flower opening. Another group of genes that continuously decreased from the S1 to S2 stage may regulate cell proliferation.

Subsequently, the expression patterns of five petal-associated and two lip-associated *TIFY* genes were analyzed (Figure 7). Transcript levels of *PaJAZ9* and *PaJAZ11* continued to decrease with petal growth in *Phalaenopsis*. Guan et al. [17] isolated a TIFY family gene *CmJAZ1-like* that was continuously downregulated with the gradual elongation of chrysanthemum petals and proved that it can limit petal size by repressing cell expansion. From these results, we could infer that *PaJAZ9* and *PaJAZ11* may also be involved in regulating the petal size of *Phalaenopsis* by inhibiting cell expansion. In addition, *PaJAZ1*, *PaJAZ7*, and *PaZML3* were abundantly expressed in the petals during the S2 and S3 stages, indicating that they may play an important role in the rapid expansion of petals. Notably, we found that *PaPPD* and *PaZML1* were highly expressed in the lips, which are the crucial components of the unique flower type for orchid plants, revealing that these two genes may be closely related to morphological diversification for developing lip structure. Furthermore, all 13 *TIFY* genes showed lower expression levels at the S4 stage in fully opened flowers, suggesting that they might not function in preventing flower abscission. In summary, these data provide a basis for exploring the role of *TIFY* genes in mediating *Phalaenopsis* flower development.

## 4. Materials and Methods

### 4.1. Identification and Physicochemical Properties Analysis of TIFY Family Genes

The whole genome data of *P. aphrodite* [27] were downloaded to identify the TIFY family genes. We used the TIFY protein sequences of 18 *A. thaliana* and 20 *O. sativa* as queries acquired from TAIR (http://www.arabidopsis.org, accessed on 24 October 2023) and TIGR (http://rice.plantbiology.msu.edu, accessed on 24 October 2023) databases (Appendix A). First, a local BLASTp search was performed using TB tools (http://cj-chen.github.io/tbtools), which is a toolkit for biologists integrating various biological data-handling tools with a user-friendly interface [38]. The conserved TIFY domain (PF06200) was downloaded from the Pfam database (http://pfam.xfam.org/, accessed on 6 December 2023) to perform an HMMER search using TBtools [38]. Considering the results of BLAST and Hmmsearch, putative TIFY proteins were further examined using NCBI Batch–CDD tools (https://www.ncbi.nlm.nih.gov/Structure/bwrpsb/bwrpsb.cgi, accessed on 8 December 2023). Finally, proteins containing the complete TIFY domain were retained for sequence analysis (Appendix A). The physicochemical properties of *Phalaenopsis TIFY* genes were obtained using TBtools [38], and their subcellular localization was predicted using the online website Cell-PLoc (http://www.csbio.sjtu.edu.cn/bioinf/Cell-PLoc/, accessed on 8 December 2023).

### 4.2. Phylogenetic Analysis

The TIFY protein sequences of *A. thaliana* and *O. sativa* were obtained from the TAIR and TIGR databases, as described above, and together with the amino acid sequences of *Phalaenopsis* TIFY proteins, they were imported into MEGA software (7.0) [39]. All TIFY proteins in the three species were aligned using MUSCLE [40]. A phylogenetic analysis was performed using the neighbor-joining method based on the best model and implemented with 1000 bootstrap replicates. Finally, the phylogenetic tree was visualized and decorated using EvolView [41].

### 4.3. Gene Structure and Conservative Domain Analysis

Based on the genome gff data, the intron–exon structures of 19 *TIFY* genes were analyzed using TBtools [38]. Then, the MEME website (https://meme-suite.org/meme/doc/meme.html, accessed on 10 December 2023) was used to explore the conserved motifs of TIFY proteins and ten motifs were assigned as upper bounds. As described above, the protein sequences were submitted to NCBI Batch–CDD tools for conserved domain analysis to obtain the hit data. Moreover, the gene structure and conserved domains were visualized using TB tools [38].

### 4.4. Prediction of cis-Acting Elements

To investigate the putative *cis*-acting elements in the promoter of *Phalaenopsis TIFY* genes, upstream 2000 bp sequences were extracted from the *Phalaenopsis* genome using TBtools. Subsequently, the PlantCARE database (http://bioinformatics.psb.ugent.be/webtools/plantcare/html/, accessed on 10 December 2023) was used to predict the *cis*-acting elements in the upstream regions of *Phalaenopsis TIFY* genes. TBtools was used to visualize *cis*-acting element functions and numbers [38].

### 4.5. Gene Duplication and Synteny Analyses

Gene duplication and synteny analyses of TIFY family members were performed using the one-step MCScanX function in the TBtools v1.120 software. The advanced circle function was adopted to visualize the chromosome distribution and segmental duplication events of the 19 *TIFY* genes in *Phalaenopsis*. Moreover, the homology of the *TIFY* genes between *Phalaenopsis* and the other two representative species (*A. thaliana* and *O. sativa*) was determined using a Dual Synteny Plot in TBtools. The genome fasta and gff3 files of *Arabidopsis* and rice were obtained from the NCBI website (https://www.ncbi.nlm.nih.gov/genome, accessed on 24 October 2023). The Simple *Ka/Ks* Calculator function in TBtools was employed to calculate the four segment-duplicated gene pairs’ Ka, Ks, and Ka/Ks values.

### 4.6. Expression Analysis

The RNA-seq expression profiles of *PaTIFYs* were mined from the transcriptome data by Chao et al. [27]. Transcript quantification and fragment per kilobase of transcript per million mapped read (FPKM) values for each gene were calculated using RSEM [42]. Expression heat maps were generated using TBTools [38].

Total flowers, petals, and lips at the four flower-opening stages of *Phalaenopsis* were harvested for the expression analysis of *TIFY* genes, and each sample had three replicates. Uniform *Phalaenopsis* plants were obtained from the Forest Orchid Garden of Fujian Agriculture and Forestry University. Total RNA was extracted from each sample using an RNA Isolation Kit (Waryong). Subsequently, 0.5 µg of total RNA was reverse transcribed into cDNA as a template for qRT-PCR. qRT-PCR was performed as described by Guan et al. [16]. The 2×RealStar Fast SYBR qPCR Mix and 96-well plates used in this process were purchased from GenStar (Beijing, China). Gene-specific primers were designed using the NCBI primer design tool (https://www.ncbi.nlm.nih.gov/tools/primerblast/, accessed on 24 December 2023). *Phalaenopsis ACTIN* (*PaACT*) was the normalization control [43]. All the primers used are listed in Appendix A. Each sample contained three biological and three technical replicates, and the relative expression levels of transcripts were calculated via the 2^−ΔΔCt^ method [44].

## 5. Conclusions

TIFY family genes are known to play an essential role in various processes of plant growth and development. However, the identification and functional analysis of TIFY proteins remain unexplored in Orchidaceae, except *D. officinale*. In this study, nineteen putative TIFY genes were identified and classified into four clades by phylogenetic analysis with more members discovered in the JAZ subfamily. Motif, gene structure, and conserved domain revealed that Phalaenopsis TIFY genes in the same subfamily/subgroup are conserved. Promoter cis-elements analysis indicated that PaTIFYs are regulated by light and multiple phytohormones. In addition, 13 TIFY genes highly expressed during flower opening exhibited two types of expression trends at four flower opening stages based on the transcriptome data and qRT-PCR validation, suggesting their different roles in cell proliferation and cell expansion of floral organ growth during Phalaenopsis flower opening. Our study provides a comprehensive analysis for exploring the evolutionary and biological functions of TIFY proteins in Phalaenopsis. In particular, our study builds a foundation for further uncovering the role of TIFY genes in mediating the late stage of Phalaenopsis flower development, which will provide some clues for improving the ornamental quality of *Phalaenopsis* in the future.

## Figures and Tables

**Figure 1 ijms-25-05422-f001:**
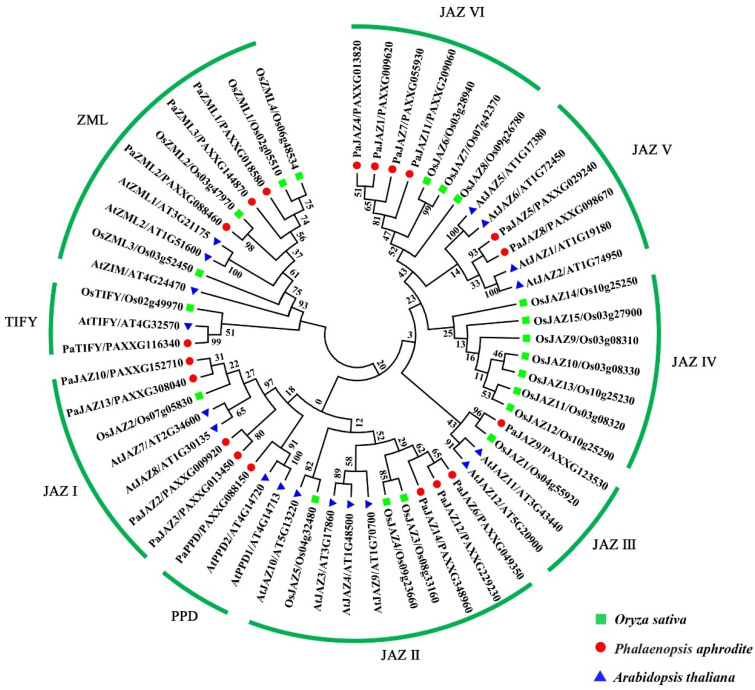
Phylogenetic tree of TIFY proteins from *Phalaenopsis*, *Arabidopsis*, and *Oryza sativa* (rice) using the neighbor-joining method with MEGA 7.0 software.

**Figure 2 ijms-25-05422-f002:**
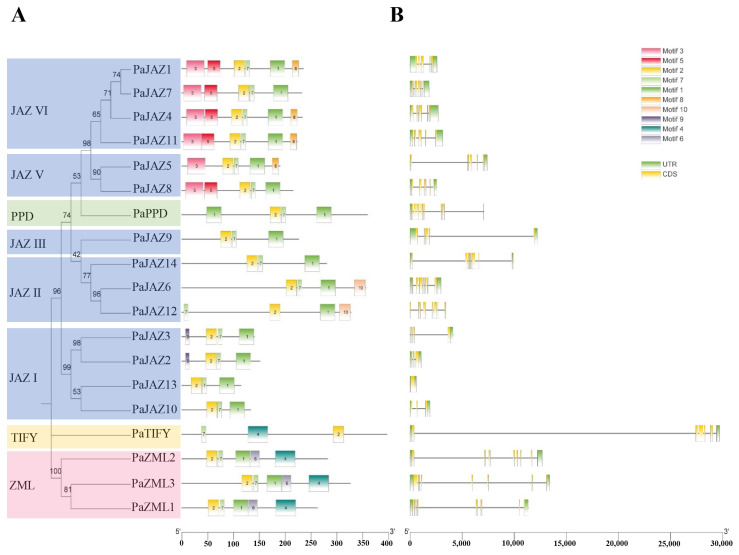
The gene structure and conserved motifs of the *Phalaenopsis* TIFY gene family. (**A**) Predicted motifs with the phylogenetic tree of *Phalaenopsis* TIFYs. (**B**) The exon–intron structure of *Phalaenopsis TIFY* genes. Yellow boxes represent exons, gray lines represent introns, and green boxes indicate upstream or downstream untranslated regions (UTR).

**Figure 3 ijms-25-05422-f003:**
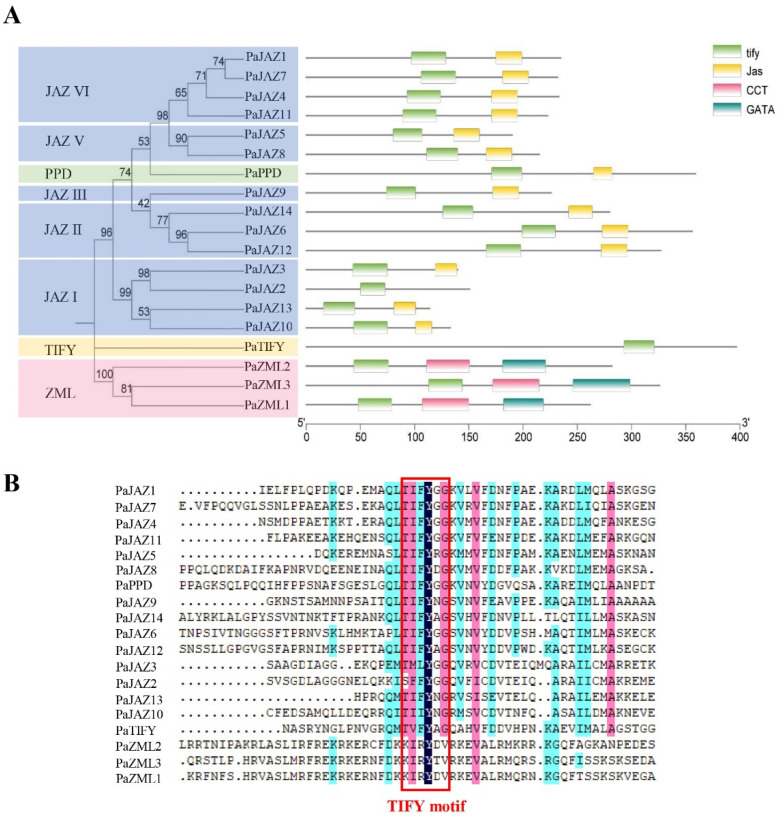
Conserved domain distributions of *Phalaenopsis* TIFY gene family and polypeptide sequence alignment of their TIFY domain. (**A**) The distribution of the conserved domains of *Phalaenopsis* TIFY proteins. Different colored blocks represent different conserved domains. (**B**) Polypeptide sequence alignment of the TIFY domain in *Phalaenopsis*. Navy blue indicates 100% identity, pink indicates 75% identity, and light blue indicates 50% identity.

**Figure 4 ijms-25-05422-f004:**
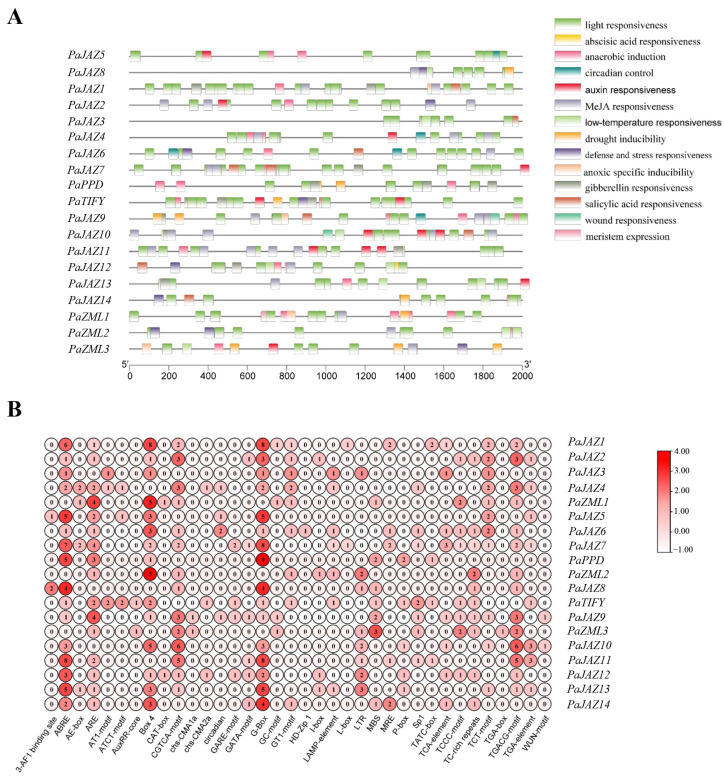
*Cis*-acting elements in the *TIFY* promoter region. (**A**) The distribution and functional classification of *cis*-acting elements 2000 bp upstream from the *TIFYs*. (**B**) The numbers of each *cis*-acting element in the *TIFY* gene promoter.

**Figure 5 ijms-25-05422-f005:**
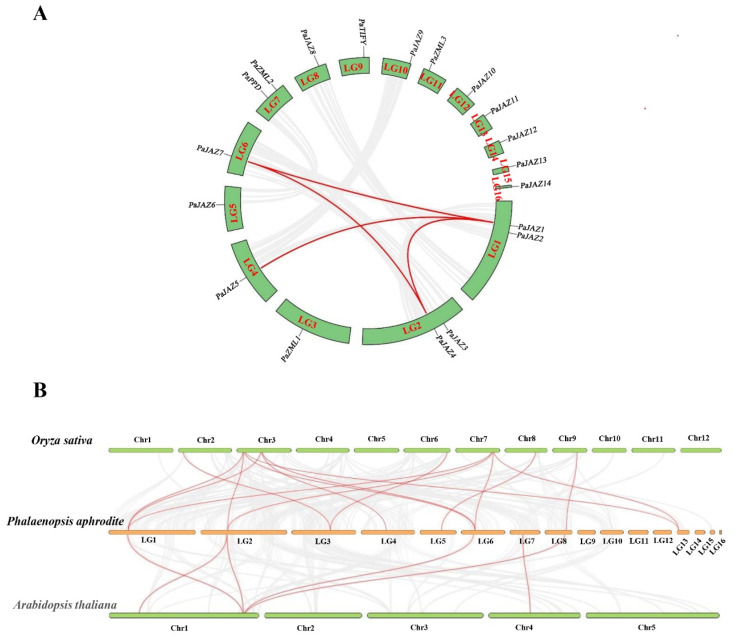
Synteny analysis within and between species of *TIFY* genes. (**A**) Chromosomal distribution and synteny analysis of *Phalaenopsis TIFY* genes, where the red lines indicate segmentally duplicated gene pairs. (**B**) Collinearity analysis of *TIFY* genes between *Phalaenopsis* and two representative plant species (*O. sativa* and *A. thaliana*), where the red lines represent syntenic *TIFY* gene pairs.

**Figure 6 ijms-25-05422-f006:**
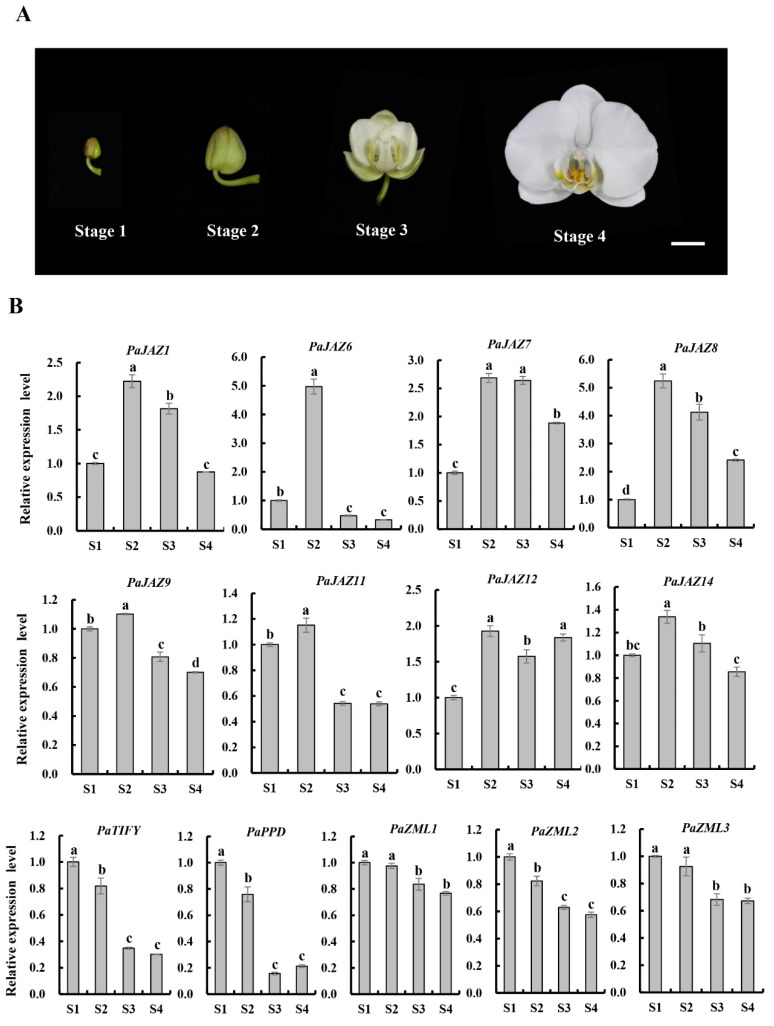
The transcription levels of *Phalaenopsis TIFY* genes during flower opening. (**A**) Four stages of *Phalaenopsis* during flower opening, scale bar = 2 cm. (**B**) Expression profiles of 13 *TIFY* genes at four stages of flower opening via qRT-PCR. The values are the mean ± SE (*n* = 3). Significant differences were examined by Duncan’s multiple-range test (Different letters above the bars indicate *p* < 0.05).

**Figure 7 ijms-25-05422-f007:**
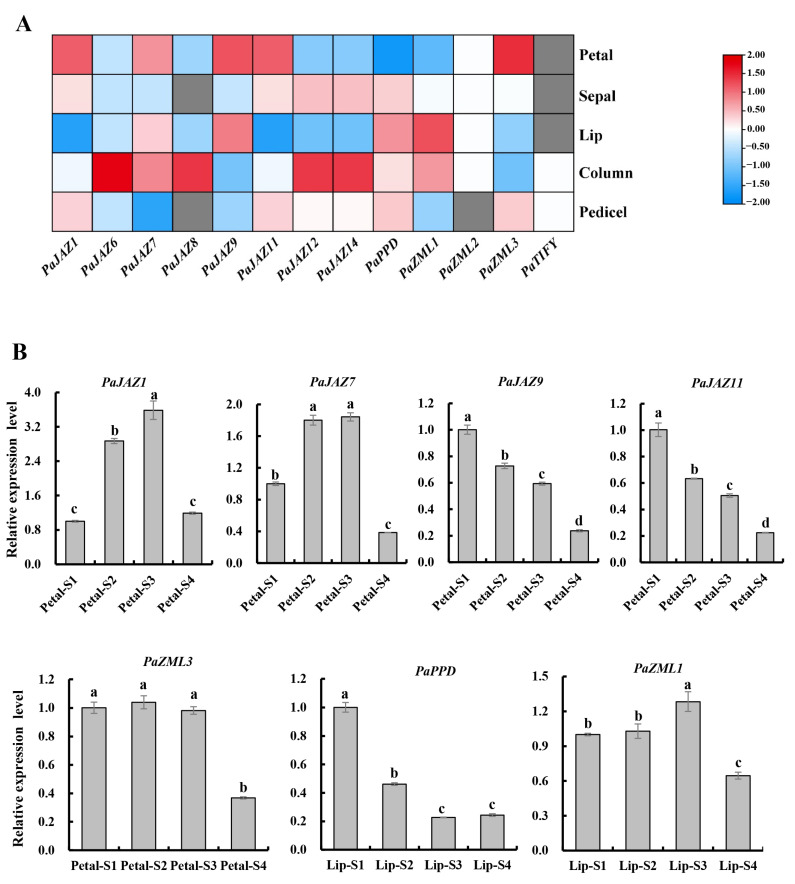
The expression level analysis of *TIFY* genes in petals or lips at four flower opening stages. (**A**) Heatmap of 13 *TIFY* genes in petals, sepals, lips, columns, and pedicels of *Phalaenopsis*. Red rectangles indicate high expression, while the blue rectangles represent low expression. (**B**) qRT-PCR detection of strongly petal- or lip-transcribed *TIFY* genes at four stages of flower opening. The values are the mean ± SE (*n* = 3). Significant differences were examined by Duncan’s multiple-range test (Different letters above the bars indicate *p* < 0.05).

**Table 1 ijms-25-05422-t001:** Physiochemical parameters and subcellular location of the 19 *TIFY* genes in *Phalaenopsis*.

Gene Name	Gene ID	ORF (bp)	AA (aa)	pI	Mw (KDa)	Instability Index	GRAVY	Subcellular Localization
*PaJAZ1*	PAXXG009620	705	234	8.76	25.29	55.48	−0.419	nucleus
*PaJAZ2*	PAXXG009920	453	150	8.24	16.3	96	−0.597	nucleus
*PaJAZ3*	PAXXG013450	420	139	8.93	15.32	65.03	−0.453	nucleus
*PaJAZ4*	PAXXG013820	699	232	8.49	25.38	56.71	−0.526	nucleus
*PaJAZ5*	PAXXG029240	570	189	9.47	21.69	49.29	−0.877	nucleus
*PaJAZ6*	PAXXG049350	1068	355	8.64	38.38	48.39	−0.412	nucleus
*PaJAZ7*	PAXXG055930	696	231	8.69	24.82	47.29	−0.428	nucleus
*PaJAZ8*	PAXXG098670	645	214	7.7	24.35	50.81	−0.774	nucleus
*PaJAZ9*	PAXXG123530	678	225	6	23.85	67.31	−0.249	nucleus
*PaJAZ10*	PAXXG152710	399	132	5.23	14.9	44.26	−0.404	nucleus
*PaJAZ11*	PAXXG209060	669	222	7.74	24.35	35.93	−0.491	nucleus
*PaJAZ12*	PAXXG229230	981	326	9.23	34.93	58.72	−0.38	nucleus
*PaJAZ13*	PAXXG308040	342	113	9.51	12.74	75.96	−0.658	nucleus
*PaJAZ14*	PAXXG348960	840	279	9.72	30.85	46.82	−0.501	nucleus
*PaPPD*	PAXXG088150	1077	358	9.36	38.87	68.71	−0.638	nucleus
*PaZML1*	PAXXG018580	786	261	8.59	27.98	49.83	−0.544	nucleus
*PaZML2*	PAXXG088460	846	281	5.61	30.41	53.62	−0.594	nucleus
*PaZML3*	PAXXG144870	978	325	6.06	35.05	35.81	−0.668	nucleus
*PaTIFY*	PAXXG116340	1191	396	8.41	41.7	52.86	−0.337	nucleus

## Data Availability

The whole genome data of *P. aphrodite* were downloaded from NCBI BioProject database (https://www.ncbi.nlm.nih.gov/bioproject/, accessed on 24 October 2023) under accession number PRJNA383284. The sequence data used in the study can be found in the Appendix A. Furthermore, the TIFY protein sequences of 18 *A. thaliana* and 20 *O. sativa* were acquired from TAIR (http://www.arabidopsis.org, accessed on 24 October 2023) and TIGR (http://rice.plantbiology.msu.edu, accessed on 24 October 2023) databases.

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
