# Peer review of "Genome-Wide Identification and Expression Pattern Analysis of TIFY Family Genes Reveal Their Potential Roles in Phalaenopsis aphrodite Flower Opening"

_ijms, 2024, doi:10.3390/ijms25105422_

Round 1

Reviewer 1 Report (Previous Reviewer 1)

Comments and Suggestions for Authors

Most of my comments have been addressed, there is no more comments about it. But there is no experiment to explore TIFY genes function, which may be concerning in future.

Author Response

Response: Thank you. we are much grateful to your carefully review and putting forward the valuable suggestions which are of great help for the improvement of our manuscript. We will focus on exploring the function of TIFY genes in Phalaenopsis flower opening in the future.

Reviewer 2 Report (New Reviewer)

Comments and Suggestions for Authors

Reviewer’s Comment / Report

The manuscript #ijms-2989794 entitled “Genome-wide identification and expression pattern analysis of TIFY family genes reveal their potential roles in Phalaenopsis aphrodite flower opening" has been reviewed.

The authors have analyzed 13 candidate TIFY genes screened from the transcriptome data exhibited two types of expression trends, suggesting their different roles in cell proliferation and cell expansion of floral organ growth during Phalaenopsis flower opening.

The author reports in the manuscript that in the study TIFY protein sequences of 18 A. thaliana and 20 O. sativa as queries acquired from TAIR (http://www.arabidopsis.org) and TIGR (http://rice.plantbiology.msu.edu) databases and analyzed.

However, the sequence information is entirely missing from the manuscript and the suppl. Files are not available to validate the experimental results

There is no proper information for sequence mapping for different gene categories.

Author Response

Response: Thank you, we appreciate this comment. We have provided the TIFY protein sequences of 18 A. thaliana and 20 O. sativa which obtained from TAIR (http://www.arabidopsis.org) and TIGR (http://rice.plantbiology.msu.edu) in Table S1. Please check it.

Round 2

Reviewer 2 Report (New Reviewer)

Comments and Suggestions for Authors The manuscript #ijms-2989794 entitled “Genome-wide identification and expression pattern analysis of TIFY family genes reveal their potential roles in Phalaenopsis aphrodite flower opening" has been reviewed. The authors have analyzed 13 candidate TIFY genes screened from the transcriptome data exhibited two types of expression trends, suggesting their different roles in cell proliferation and cell expansion of floral organ growth during Phalaenopsis flower opening. Numerous studies have confirmed that TIFY proteins are implicated in various processes of plant growth and development, in which particularly author cited various studies in page 2 line 56-70. Numerous studies have been conducted on floral organ identity, and specific flower development models. Nevertheless, little is known about the genetic regulation of the late stage of flower development in orchids. This study provides a comprehensive basis for further investigation into the underlying roles of TIFY genes in floral organ growth in Phalaenopsis. However, the study failed to confirm the different expression of these genes in other plant organs or tissue during the flowering.

Author Response

Thank you, we appreciate this comment. In this manuscript, we aimed to explore the potential functions of PaTIFYs in the late stage of flower development (i.e. flower opening process) in Phalaenopsis, so we only focused on detecting their expression patterns in floral organs. Perhaps these TIFY genes also play a role in the development of other tissues, which requires further exploration in future research.

This manuscript is a resubmission of an earlier submission. The following is a list of the peer review reports and author responses from that submission.

Round 1

Reviewer 1 Report

Comments and Suggestions for Authors

 In the manuscript named “Genome-Wide Identification and Expression Pattern Analysis of TIFY Family Genes Reveal their Potential Roles in Phalaenopsis Flower Opening”, Yunxiao Guan et al have performed genome-wide analysis of TIFY genes in Phalaenopsis, they have identified 19 gene members, and characterized their gene structures, evolution, and promoters. In addition, they have also evaluated their expressional profiles in flower development and opening process. Their findings were helpful for determining TIFY genes functions in Phalaenopsis, which would be applicated in future. However, there are some comments about this research.

1. Most works were performed in silico, just qRT-PCR was wet experiment, more molecular experiments were needed to explore these genes functions, such as GFP, transgene expressional analysis in model plants, etc.

2. The genes have termination codons, which have no amino acids in protein production, therefore, the lengths of protein may shorter than ORF/3, please confirm table 1 and the text.

3. All TIFY gene should be renamed based their homologous in Arabidopsis and rice, which is easily for distinguishing TIFY members, didn’t use locus id in analysis flow, please correct them in manuscript and figures.

4. Figure 2B was ugly, please don’t too excessively stretch the figure.

5. The figure 5B and 5C should be merge into one figure, the synteny analysis could be performed with three species, Arabidopsis, rice, and Phalaenopsis in one analysis event, then merge them into one figure.

6. How authors select 13 TIFY genes in expressional analysis? qRT-PCR have evaluated 12 genes in figure 6, while six genes in figure 7. In addition, these genes weren’t detected in all samples, why? Please add comments or descriptions in manuscript, for example, in method section.

7. The discussion should be described in detail, especially more words should be organized with your findings or results.

Reviewer 2 Report

Comments and Suggestions for Authors

Comments and Suggestions:

  1. Acceptance with Minor Revision: Overall, this manuscript provides a comprehensive analysis of the TIFY gene family in Phalaenopsis flower opening, filling a significant gap in our understanding of orchid flower development. The findings are substantial and contribute significantly to the field of plant molecular biology. I recommend accepting this manuscript with minor revisions.

  2.  

  3. Clarifications:

    • Could you elaborate on the methodology used for transcriptome data analysis and how the expression profiles of the TIFY genes were determined?
    •  
    • It would be beneficial to provide more details on the criteria used for selecting the 13 candidate TIFY genes from the transcriptome data.
    • Please clarify how the expression patterns of the TIFY genes were validated through qRT-PCR experiments. Was there a specific housekeeping gene used for normalization?
  4.  
    • *Line 122: "the online website MEME" - Consider revising to "the online tool MEME" for smoother phrasing.
      1. *Line 217: "15 and six" could be clearer as "15 with rice and six with Arabidopsis" for better clarity.
    • Line 220: "monocot O. sativa" should be corrected to "monocot rice, O. sativa"
    • *Line 221: "and have" should be "and share".
    •  

    •  
    •  
    •  
  5.  
  6. Discussion Points:

    • It would be insightful to discuss the potential functional roles of the TIFY genes identified in Phalaenopsis flower opening in comparison to other plant species, especially considering the unique floral morphology of orchids.
    •  
    • Considering the involvement of TIFY genes in various plant processes, including hormone signaling and stress responses, could you speculate on their potential roles beyond flower opening in Phalaenopsis, such as in response to environmental cues or interactions with pollinators?
    •  
  7. Conclusion Refinement: The conclusion provides a succinct summary of the findings. However, it would be beneficial to emphasize the significance of the study's implications for orchid flower development and potential applications in breeding and horticulture.

Overall, this manuscript presents valuable insights into the TIFY gene family's involvement in Phalaenopsis flower opening, and addressing the minor revisions and clarifications will further enhance the manuscript's quality and impact.

Comments on the Quality of English Language
  1. English Language Revision: While the content of the manuscript is strong, I suggest having a native English speaker or a professional editor review the text for any grammatical errors or awkward phrasing.

Reviewer 3 Report

Comments and Suggestions for Authors

The article deals with important TIFY genes that are unique to plants and have been studied in only a few species. In addition, flower development in Orchidaceae is now being intensively studied (e.g., https://www.frontiersin.org/journals/plant-science/articles/10.3389/fpls.2022.901089/full ), so the article is part of this scientific trend.

Editorially and linguistically, the article is correct.

The methods are properly applied. Also the conclusions are correct and reflect the most important results of the work.

I believe that the main authors' achievement is to link the expression of TIFY genes to the different stages of flower development in Phalaenopsis aphrodite.

Critical comments:

The article would have benefited greatly if the authors had analyzed at least two species in the genus Phalaenopsis.

Please add the full species name to the title, because now the title is misleading.

Why did the authors only use proteins from Phalaenopsis, Arabidopsis, and Oryza sativa in the Phylogenetic tree of TIFY ? It would be appropriate to improve this tree using data obtained from still other  species e.g. Triticum aestivum and Dendrobium (see Ebel C, BenFeki A, Hanin M, Solano R, Chini A. Characterization of wheat (Triticum aestivum) TIFY family and role of Triticum Durum TdTIFY11 a in salt stress tolerance. PLoS ONE. 2018;13(7):18. ). What are the differences between Dendrobium and Phalaenopsis ?

Minor note: Please for the article how the whole flowering plant looks like.